# DEPN: Detecting and Editing Privacy Neurons in Pretrained Language Models

**Xinwei Wu[1], Junzhuo Li[2], Minghui Xu[1], Weilong Dong[1],**
**Shuangzhi Wu[3], Chao Bian[3,4], Deyi Xiong[1,2]\***

[1]College of Intelligence and Computing, Tianjin University, Tianjin, China
[2]School of New Media and Communication, Tianjin University, Tianjin, China
[3]Department of Computer Science and Technology, Tsinghua University, Beijing, China
[4]ByteDance Lark AI, Beijing, China
{wuxw2021,jzli,xuminghui,willowd,dyxiong}@tju.edu.cn,
wufurui@bytedance.com, bianc18@mails.tsinghua.edu.cn

## Abstract

Large language models pretrained on a huge amount of data capture rich knowledge and information in the training data. The ability of data memorization and regurgitation in pretrained language models, revealed in previous studies, brings the risk of data leakage. In order to effectively reduce these risks, we propose a framework DEPN to Detect and Edit Privacy Neurons in pretrained language models, partially inspired by knowledge neurons and model editing. In DEPN, we introduce a novel method, termed as privacy neuron detector, to locate neurons associated with private information, and then edit these detected privacy neurons by setting their activations to zero. Furthermore, we propose a privacy neuron aggregator dememorize private information in a batch processing manner. Experimental results show that our method can significantly and efficiently reduce the exposure of private data leakage without deteriorating the performance of the model. Additionally, we empirically demonstrate the relationship between model memorization and privacy neurons, from multiple perspectives, including model size, training time, prompts, privacy neuron distribution, illustrating the robustness of our approach.

## 1 Introduction

Remarkable progress has been made in large language models (LLMs) in recent years (Brown et al., 2020; Liu et al., 2021; Ouyang et al., 2022; Lee et al., 2023). However, despite this success, LLMs are confronted with privacy and security concerns in real-world applications (Guo et al., 2022; Brown et al., 2022; Li et al., 2023). The primary cause of privacy and security risks is the inherent nature of large pretrained language models. Previous studies (Carlini et al., 2019, 2021; Thakkar et al., 2021; Henderson et al., 2018) have demonstrated that pretrained language models tend to memorize and regurgitate a significant portion of the training data, including atypical data points that appear only once in the training data. Additionally, external factors (e.g., membership attack) also contribute to these risks. A variety of methods have been explored to attack LLMs for training data extraction. For instance, Carlini et al. (2021) have successfully extracted personal information from GPT-3's output, while Li et al. (2023) have induced the generation of personal information by utilizing multi-step prompts in ChatGPT. All these show that large pretrained language models suffer from a serious risk of privacy leakage.

In order to safeguard privacy, numerous methods have been proposed. The majority focus on either removing sensitive information during the data processing stage (Liu et al., 2017; El Emam et al., 2009; Zhou et al., 2008; García-Pablos et al., 2020), or reducing the extent to which models memorize training data during the training stage (Li et al., 2021; Hoory et al., 2021; Plant et al., 2021; Coavoux et al., 2018). However, privacy breaches often come to light after the completion of model training, rendering previous methods less effective. There are also methods proposed in the post-processing stage, which involve slight parameter retraining to make the model forget privacy information (Bourtoule et al., 2021; Gupta et al., 2021; Neel et al., 2020). Nevertheless, these methods generally incur high computational complexity, making it challenging to apply them to complex model architectures. In practice, model developers often attempt to prevent language models from outputting specific information via blocking or filtering certain keywords, which, however, does not truly address the underlying issue.

We speculate that private information might be

---

*Corresponding author.

stored in specific neurons, just like knowledge neurons (Geva et al., 2021; Meng et al., 2022; Dai et al., 2022). This presumption suggests that we could change the model memorization of private information by detecting and deleting these neurons (termed as privacy neurons). Therefore, we propose a framework DEPN for detecting and editing privacy neurons. To detect privacy neurons, we introduce a privacy neuron detector that uses gradient integration to simultaneously compute the contributions of multiple markers to neuron activations. This allows us to estimate an overall privacy attribution score for private information. Subsequently, we further propose a privacy neuron editor that simply sets the activations of the top $z$ privacy neurons with the highest privacy scores to zero to erase the model memorization of the corresponding private information. For the scenario of processing multiple sentences at the same time, we also present a privacy neuron aggregator to facilitate privacy information editing in batches.

Experimental results show that our framework can quickly reduce the risk of private data leakage without affecting model performance. Compared with other methods, our framework is highly efficient. Furthermore, we have found that model memorization leads to the aggregation of privacy neurons in our experiments, and demonstrated that our framework is very suitable for the scenario of deep model dememorization.

The main contributions of our work are summarized as follows:

- For the first time, we explore model editing into privacy protection of pretrained language models, provide a new way for privacy protection, and propose DEPN to effectively eliminate model memorization in the post-processing stage.

- We propose the privacy neuron detector to localize privacy neurons based on gradient attribution, and the privacy neuron editor to dememorize privacy information in pretrained language models.

- We conduct experiments to demonstrate that the proposed framework is capable of protecting privacy leakage from pretrained language models.

## 2 Preliminary

**Privacy Definition**   Privacy preservation has become an issue of great concern in the era of pre-trained language models. Protecting privacy first requires specifying the boundaries of privacy. The definition of privacy is broad. It is closely related to its context and discourse (Brown et al., 2022). In any texts about, a specific person can be considered as private. For the convenience of research, a narrow definition of privacy is usually taken (Sousa and Kern, 2023), which treats personal identity information as privacy, such as names, ID numbers, phone numbers and other related expressions. The proposed DEPN can be adapted to the above two definitions.

**Model Editing**   Geva et al. (2021) find that the feed-forward network module in Transformer (i.e., a two-layer perceptron) can be considered as a key-value memory, where each key corresponds to a text pattern and each value represents a distribution over the vocabulary. Based on this finding, a strand of research, (Geva et al., 2021; Meng et al., 2022; Dai et al., 2022) propose to edit factual knowledge encoded in pre-trained LLMs by locating neurons related to the entities of factual knowledge. The basic idea of localization is to change the parameters of neurons, and then observe the changes in the probability of the object entity predicted by the model. The neurons with greater influence on the probability are more closely related to the object entity.

However, these methods have a limitation that they can only observe the probability change of one token at a time. Semantic units are usually composed of a sequence of tokens, rather than a single token. This makes it impossible to use these methods directly.

## 3 Methodology

The proposed DEPN consists of three components: the privacy neuron detector (§3.2), the privacy neuron editor (§3.3) to erase the model memorization of privacy data, and the privacy neuron aggregator (§3.4) for privacy preservation in batches.

### 3.1 Privacy Prediction Task

Given a tuple $\boldsymbol{T} = \{\boldsymbol{X}, \boldsymbol{Y}\}$, let $\boldsymbol{Y} = \{y_1, ..., y_n\}$ be the sequence with private information, $\boldsymbol{X}$ be the the context of the sequence, $\boldsymbol{\theta}$ be the parameters of a language model. Given a context $\boldsymbol{X}$, the

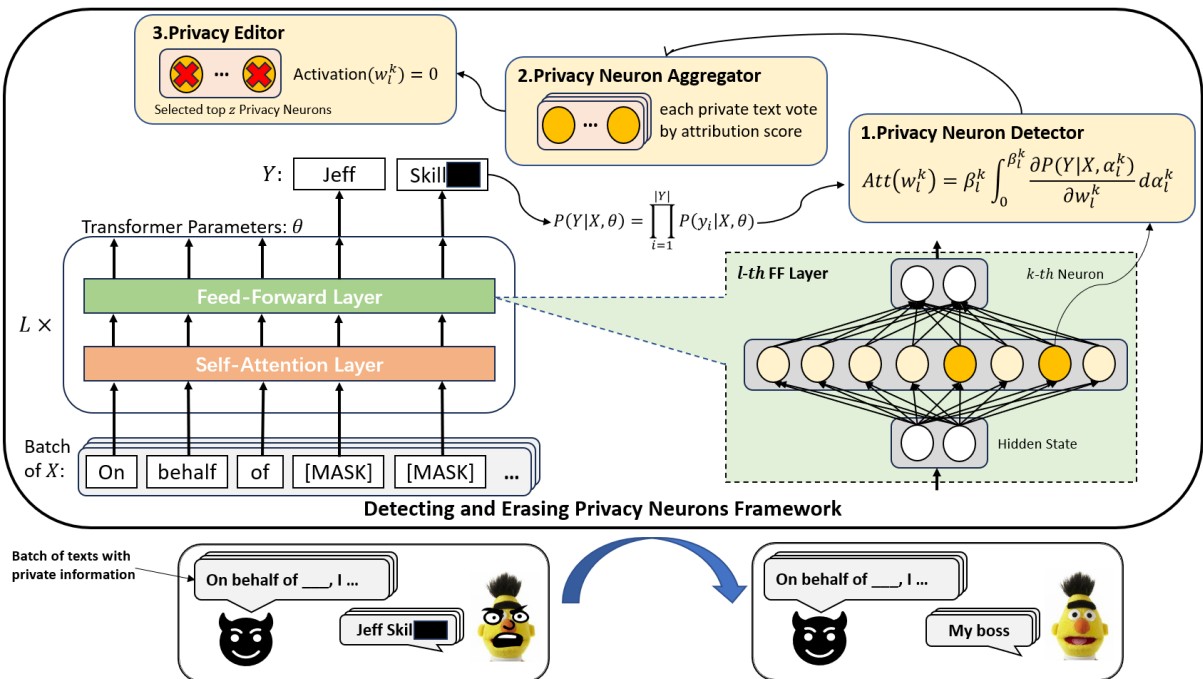

Figure 1: The diagram of DEPN. When a language model leaks privacy information, DEPN calculates privacy attribution scores using the Privacy Neuron Detector. It then selects the top $z$ privacy neurons with the Privacy Neuron Aggregator and eliminates the model memorization of privacy information using the Privacy Editor.

probability of the language model yielding a token is $P(y_i|\boldsymbol{X}, \boldsymbol{\theta}), y_i \in \boldsymbol{Y}$, so the probability of the model leaking the private sequence is:

$$P(\boldsymbol{Y}|\boldsymbol{X}, \boldsymbol{\theta}) = \prod_{i=1}^{|\boldsymbol{Y}|} P(y_i|\boldsymbol{X}, \boldsymbol{\theta}) \qquad (1)$$

Take "An■ Ka■ is a senior writer at ESPN.com" as private sentence containing a person's name "An■ Ka■". Suppose the input to the language model is "_ _ is a senior writer at ESPN.com", our goal is to reduce the probability of privacy leakage, i.e., minimizing the probability of predicting "An■" and "Ka■" .

## 3.2 Privacy Neuron Detector

As described in Section 2 factual knowledge is found to be stored in the feed-forward networks of Transformer, in the form of key-value memory. Inspired by this, we speculate that private information might be also encoded in specific neurons. Model editing has offered methods to locate and edit knowledge-related neurons. However, existing methods can only deal with semantic units composed of a single token, making them not directly applicable to detect and edit mutli-token private sequences. To address this issue, we propose a privacy attribution method based on gradient integration. The proposed privacy attribution can evaluate

which neurons play a key role in the leakage of private information from language models.

Let $w_l^k$ be a neuron to be evaluated by the privacy attribution method, where $l$ is the layer of the neuron in the language model, and $k$ is its position. According to §3.1, the probability of the model outputting private information is:

$$P(\boldsymbol{Y}|\boldsymbol{X}, w_l^k) = \prod_{i=1}^{|\boldsymbol{Y}|} P(y_i|\boldsymbol{X}, w_l^k = \alpha_l^k) \qquad (2)$$

where $\alpha_l^k$ represents the value of the $k$-th neuron in the $l$-ith FFN layer.

We gradually change the parameter of the target neuron from $0$ to the original value of the neuron. In this process, the probability of the output will accordingly change. We calculate the cumulative gradient of the probability change during this process as the neuron's contribution (i.e., privacy attribution score) to the privacy-sensitive output. The privacy attribution score is computed as:

$$\text{Att}(w_l^k) = \beta_l^k \int_0^{\beta_l^k} \frac{\partial P(\boldsymbol{Y}|\boldsymbol{X}, \alpha_l^k)}{\partial w_l^k} d\alpha_l^k \qquad (3)$$

where $\beta_l^k$ is the original value of the neuron $w_l^k$, $\frac{\partial P(\boldsymbol{Y}|\boldsymbol{X}, \alpha_l^k)}{\partial w_l^k}$ calculates the gradient of the model

output with regard to $w_l^k$. Directly calculating continuous integrals is intractable. We follow Dai et al. (2022) to use Riemann approximation:

$$\text{Att}(w_l^k) = \frac{\beta_l^k}{m} \sum_{j=1}^{m} \frac{\partial P(\boldsymbol{Y}|\boldsymbol{X}, \frac{j}{m}\beta_l^k)}{\partial w_l^k} \quad (4)$$

where $m = 20$ is the number of approximation steps.

As $P(\boldsymbol{Y}|\boldsymbol{X}, w_l^k) = \prod_{i=1}^{|\boldsymbol{Y}|} P(y_i|\boldsymbol{X}, w_l^k = \alpha_l^k)$, we have

$$\text{Att}(w_l^k) = \sum_{i=1}^{|\boldsymbol{Y}|} \frac{\beta_l^k}{m} \sum_{j=1}^{m} \frac{\partial P(y_i|\boldsymbol{X}, \frac{j}{m}\beta_l^k)}{\partial w_l^k} \quad (5)$$

If the neuron has a great influence on the output of a private information, the gradient will be significant, and a large integration value will be obtained. Therefore, the privacy attribution score can measure the neuron's contribution to the leakage of privacy information, and the greater the privacy attribution score, the greater the privacy sensitivity of the neuron. We select neurons with the top $z$ privacy attribution score as candidates for editing.

### 3.3 Privacy Editor

After detecting the privacy neuron candidates with the privacy neuron detector, we reduce the model memorization of private information by editing. Particularly, we use a simple yet effective editing strategy: setting the parameters (activation values) of the corresponding neurons to 0, so that the information flow will not pass through these privacy neurons.

### 3.4 Privacy Neuron Aggregator

As a number of sentences in the training data of LLMs contain private information, the privacy neuron detection and editing can be done over multiple sentences in a batch processing way. To erase privacy information encoded in the language model from multiple sentences in the training data, we propose the privacy neuron aggregator. When the input is a text batch, we calculate the privacy attribution score matrix of each sequence in the batch. After the privacy attribution score calculation, we let each sequence vote for neurons according to their privacy attribution scores, and select the top $z$ neurons with the most votes. These selected neurons will be edited to erase private information. The hyperparameter $z$ is adjusted according to the model size, training epochs and other factors. More details can be found in (§5.1).

## 4 Experiments

We carried out experiments to examine the effectiveness of the proposed DEPN on a dataset containing private information.

### 4.1 Setup

**Dataset** We used the Enron dataset (Klimt and Yang, 2004). It consists of employee emails that were publicly disclosed during Enron's legal investigation by the Federal Energy Regulatory Commission. It is the largest publicly available collection of "real" email data, containing over 500,000 emails from 158 users.[1] We randomly sampled 5% of the data from Enron as the validation dataset to evaluate model performance.

**Private Information Sampling** In our study, we categorized the private information in the Enron dataset into two types: private phrases (for the narrow definition of privacy), such as names and phone numbers, and a batch of randomly sampled sentences to be edit. **Names**: We selected 20 unique names that are memorized by language models, found in 126 sentences, such as "An■ Ka■ is a senior writer at ESPN.com". **Phone Numbers**: We also selected 20 unique LM-memorized phone numbers, such as "My phone number is 7 1 3 8 5 ■ ■ ■ ■ ■". **Private texts**: We randomly selected 100 sentences that are not semantically overlapping with each other. In Appendix A.4, we discuss how we determine whether private information is memorized by a language model.

**Model Settings** We conducted experiments using the widely used pretrained model, **BERT-base** (Devlin et al., 2018). The model consists of 12 transformer layers, with a hidden state size of 768 and an internal hidden size of 3072 for the feed-forward network (FFN). Our experiments were performed on NVIDIA Tesla A6000 graphics processors. More training details are show in Appendix A.1.

**Baselines** To demonstrate the effectiveness and robustness of DEPN, we compared it with the following baseline models. **BERT-O:** The bert model that has not been trained on the Enron dataset. Since the model does not know the privacy information in the dataset, it provides an oracle for assessing the risk of privacy leakage; **BERT-F:** The

---

[1] https://www.cs.cmu.edu/~enron/

| Privacy Type | Models | Time ↓ | Valid-PPL ↓ | Privacy Leakage Risk | |
|---|---|---|---|---|---|
| | | | | Metric | Value |
| Phone Number | BERT-O | - | 25.23 | Exposure ↓ | **1.58** |
| | BERT-F | 100% | **3.07** | | 15.74 |
| | BERT-FE | **2.4%** | 3.11 | | 9.78 |
| | BERT-DP | 181.4% | 5.43 | | 3.12 |
| Name | BERT-O | - | 25.23 | MRR ↓ | **0.87** |
| | BERT-F | 100% | **3.07** | | 1.21 |
| | BERT-FE | **4.4%** | 3.11 | | 1.15 |
| | BERT-DP | 181.4% | 5.43 | | 0.95 |
| Random Text | BERT-O | - | 25.23 | PPL ↑ | **10.05** |
| | BERT-F | 100% | **3.07** | | 2.30 |
| | BERT-FE | **4.6%** | 3.11 | | 3.67 |
| | BERT-DP | 181.4% | 5.43 | | 8.82 |

Table 1: Results of testing the risks of leaking private Phone Numbers, Names, and Texts on different baseline models, as well as the efficiency of protection. **Bold** and underlined results indicate the best and second best result, respectively. ↑: the higher the better. ↓: the lower the better.

bert model trained on the Enron dataset, which corresponds to the best predictive performance on the Enron dataset, but has the greatest risk of privacy leakage; **BERT-DP:** A privacy model trained by the differential privacy gradient descent method (Li et al., 2021) on the Enron dataset, which is the commonly used privacy protection method when using private data for training.

We applied our proposed DEPN on **BERT-F** to make a safe model, which is referred to as **BERT-FE** in following experiments. Our codes are available now.[2]

**Metrics**  To observe the effect of different privacy preserving methods on the model performance, we use the Perplexity of Masked Language Modeling task on the Enron validation dataset (**Valid-PPL**) as the metric.

Due to the different types of private information, we provide metrics separately for the risk of privacy leakage.

**Exposure:** The exposure (Carlini et al., 2019) metric is commonly used in privacy attacks to measure the exposure risk of phone numbers. Given a number sequence $c$, a model with parameters $\boldsymbol{\theta}$, and the randomness space $\mathcal{R}$, the exposure $e_{\boldsymbol{\theta}}$ of $c$ can be calculated as :

$$e_{\boldsymbol{\theta}} = \log_2 |\mathcal{R}| - \log_2 \text{Rank}_{\boldsymbol{\theta}}(c). \qquad (6)$$

**Mean Reciprocal Rank (MRR):** A person's name is usually composed of multiple tokens. Therefore, we use the reciprocal average of the rank of each target token to measure the model's memorization of names. Given a prefix $\boldsymbol{Q}$, a name

[2]https://github.com/flamewei123/DEPN

token sequence $\boldsymbol{E} = \{e_1, ..., e_n\}$, the length is $|\boldsymbol{E}|$, the model predicts the rank of the target token as $rank(e_i|Q)$, and the MRR for the name $\boldsymbol{E}$ is calculated as follows:

$$\frac{\sum_{i=1}^{|\boldsymbol{E}|} \frac{1}{Rank(e_i|Q)}}{|\boldsymbol{E}|}. \qquad (7)$$

**Perplexity (PPL):** When the private text is a complete sentence, we directly use the perplexity as the measure of the model memorization.

### 4.2 Main Results

Table 1 presents our main results, including model performance, privacy leakage risk, and execution time cost. The results demonstrate the competitiveness of our framework.

For the performance on the Enron validation dataset (Valid-PPL), BERT-O, which is not trained on the Enron dataset, exhibits the poorest performance. BERT-DP trained with DP-SGD does not perform well either, due to noise introduced during backpropagation. In contrast, BERT-FE equipped with DEPN performs almost on par with BERT-F on the validation dataset, indicating that neuron erasure minimally impacts model performance.

Regarding privacy leakage risk metrics, including exposure, MRR, and PPL, clearly indicate that BERT-FE equipped with DEPN achieve the reduction of privacy leakage risk. BERT-F, trained directly on private data, exhibits the highest risk. In comparison, DEPN significantly reduces the risk of leakage. BERT-O, which has no access to private data, demonstrates the lowest risk across all three data types. The BERT-DP model also exhibits very low risk.

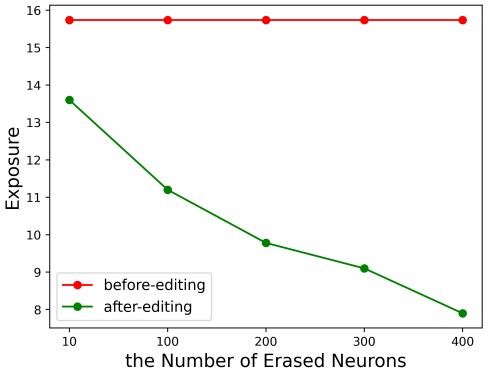
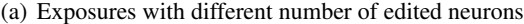

(a) Exposures with different number of edited neurons.

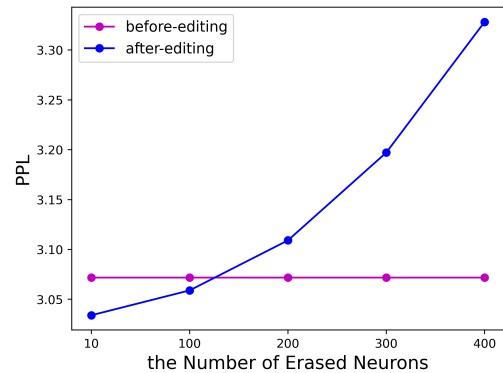

(b) Model performance with different number of edited neuron.

Figure 2: The performance of the model and the risk of privacy leakage with the change trend of the number of neurons edited.

In terms of execution time cost, we assume that the fine-tuning time of BERT-F on data excluding privacy is 100% (reference time cost). The DEPN framework requires less than 5% of the reference time cost, while BERT-DP requires more time due to gradient clipping.

In conclusion, while differential privacy training and fine-tuning with non-private data can mitigate privacy leakage risks, they incur more time and may significantly undermine model performance. The DEPN framework strikes an excellent balance between performance and privacy protection.

## 5 Analysis

We further conducted in-depth analyses to demonstrate why DEPN is able to dememorize privacy in LLMs from multiple perspectives, including analyses on the relationship between privacy neurons and model memorization, on the robustness as well as the cost-effectiveness of DEPN.

### 5.1 Effect of the Hyperparameter

Figure 2 illustrates the impact of the hyperparameter, the number of edited neurons, on the model. We calculate the exposures of the original model BERT-F and the enhanced model BERT-FE on 20 phone numbers. In Figure 2(a), the red line represents the average exposure of BERT-F, while the green line represents the average exposure of BERT-FE with varying numbers of edited neurons. As the number of edited neurons increases, the exposure significantly decreases. In Figure 2(b), the purple line represents the PPL of BERT-F on the valida-

tion set, while the blue line represents the PPL of BERT-FE on the validation set with different numbers of edited neurons. As the number of erasures increases, the PPL noticeably increases. Therefore, increasing the number of edited neurons reduces the risk of privacy leakage in the model, but it also leads to a decrease in the model performance.

### 5.2 Relationship between Memorization And Privacy Neurons

As it is widely recognized, privacy data leakage often stems from the model's ability to memorize the training data. In this subsection, we conducted experiments to investigate the relationship between model memorization and privacy neurons, providing further evidence for the effectiveness of the proposed DEPN.

**Impact of Training Time on Privacy Neuron Distribution over Layers** Figure 3 depicts the evolution of the distribution of privacy neurons over layers as the number of training epochs increases. Overall, the distribution of privacy neurons is pyramid-shaped, and most privacy neurons identified by the privacy neuron detector are located in layers 10-12 of BERT-base. Specifically, in epoch 1, about 40% of privacy neurons are in the top layer of BERT-base. As training progresses, the proportion of privacy neurons from deep layers increases to 60% by epoch 3 and to 80% by epoch 6. By the 9-th epoch, the distribution of privacy neurons remains largely unchanged compared to the 6-th epoch. This suggests that as the depth of model training increases, the memorization of

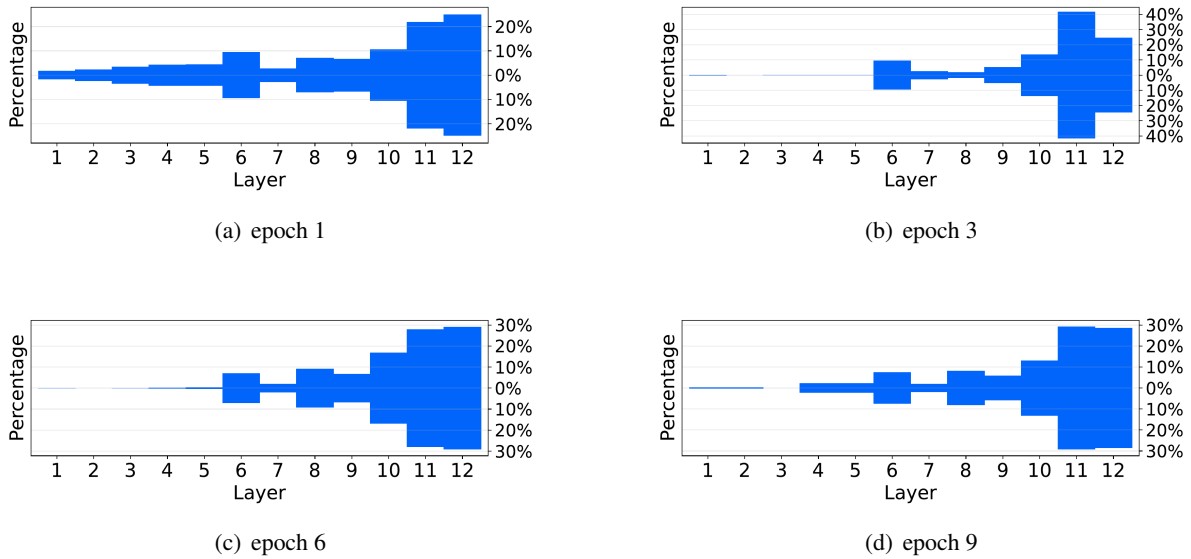

(a) epoch 1               (b) epoch 3

(c) epoch 6               (d) epoch 9

Figure 3: The distribution of privacy neurons in the bert-base model at different training epochs.

| Models | # Edited Neurons | Time | Before Editing | | After Editing | | Reduction Rate |
|--------|------------------|------|-----------|----------|-----------|----------|----------------|
| | | | Valid-PPL | Exposure | Valid-PPL | Exposure | |
| bert-small | 100 | 0.26h | 4.09 | 5.10 | 4.57 | 3.39 | 33.5% |
| bert-base | 200 | 1.59h | 3.07 | 15.74 | 3.11 | 9.78 | 37.86% |
| bert-large | 400 | 7.66h | 2.93 | 18.10 | 2.98 | 7.63 | 57.84% |

Table 2: The privacy leakage risk reduction rate for models of different sizes.

private data tends to converge.

In Appendix A.3, we conducted experiments to observe the changes of privacy leakage risk reduction at different training epoch. The results show that when the training time increases, the risk of privacy leakage increases too, and the proposed DEPN becomes more effective in privacy preservation.

**Effect of the Model Size** Table 2 illustrates the performance of the DEPN framework on models of different scales. Each model was trained for 10 epochs using the optimal hyperparameter settings. Overall, larger models require more time to identify privacy neurons and require editing a greater number of privacy neurons for optimal performance. Larger models tended to show a deeper memory for phone numbers before privacy neurons are edited, leading to higher exposure. After privacy neuron editing, from the perspective of reduction rate, the exposure of the large model is reduced even more. These findings suggest that larger models are more at risk of privacy breaches. Fortunately, the DEPN framework demonstrates better performance on larger models compared to smaller ones, offering improved protection against

privacy risks.

**Summary of the Relationship between Memorization and Privacy Neurons** Based on the aforementioned experimental findings, we can conclude that the model's scale, training time, and frequency of privacy data occurrence are all factors that have influence on the model memorization. As the model memorization of privacy data deepens, the aggregation of privacy neurons associated with privacy data becomes more pronounced, which makes the method of locating and eliminating privacy neurons more suitable for deep memorization scenarios. Therefore, the DEPN framework has demonstrated excellent effectiveness in mitigating model memorization.

### 5.3 Robustness Analysis

**Ablation Study** We conducted ablation experiments to assess the robustness of the privacy neuron detector by comparing its performance with different neuron localization methods on phone number data. In Table 4, we present the results of these experiments. Specifically, "KN" refers to the knowledge attribution approach proposed by Dai et al. (2022), while "Random" donates an approach

| Privacy Amount | # Edited Neurons | Time | Before Editing | | After Editing | |
|---|---|---|---|---|---|---|
| | | | Valid-PPL | Exposure | Valid-PPL | Exposure |
| 20 | 200 | 0.76h | 3.07 | 15.74 | 3.11 | 9.78 |
| 100 | 500 | 1.59h | 3.07 | 12.46 | 3.33 | 10.47 |
| 1000 | 2000 | 17.61h | 3.07 | 8.32 | 3.81 | 8.03 |

Table 3: Analysis results on the cost-effectiveness of DEPN.

| Methods | Before Editing | | After Editing | |
|---|---|---|---|---|
| | Valid-PPL | Exposure | Valid-PPL | Exposure |
| PND + Editing | 3.07 | 15.54 | 3.11 | **9.78** |
| KN + Editing | 3.07 | 15.54 | 3.10 | 10.75 |
| Random + Editing | 3.07 | 15.54 | 3.07 | 12.48 |

Table 4: Effect of using different neuron localization methods on results.

| Prompts | Original Exposure | Exposure |
|---|---|---|
| 'Contact me at ***' | 12.52 | 9.77 ↓ |
| 'Contact me at : ***' | 11.20 | 9.40 ↓ |
| 'Contact me : ***' | 12.50 | 9.68 ↓ |
| 'Call me at ***' | 12.31 | 11.82 ↓ |
| 'My phone number is ***' | 13.41 | 12.96 ↓ |
| 'You can call me at ***' | 13.04 | 12.84 ↓ |

Table 5: Results with varying prompts during privacy attack. 'Contact me at ***' is the prefix to the private phone numbers in the training data, and the others are varying prompts used in inference.

that randomly selects the same number of neurons as our method. Our method PND (privacy neuron detector) achieves superior performance in terms of exposure reduction compared to the other methods. Although the knowledge attribution approach gains a good exposure reduction, it is less effective than our method due to its attribution being targeted at a single token. The random selection approach is also able to decrease privacy exposure but the exposure reduction is not as significant as the KN approach and our detector. These results unequivocally demonstrate the effectiveness of our method for in privacy neuron localization.

**Robustness to Different Prompts** We conducted experiments to validate the robustness of DEPN to different prompts. We sampled private data containing phone numbers, all composed of the same prefix, from the training dataset. We then performed privacy attacks during inference using different prompts to examine whether changing prompts would still result in privacy leakage. Table 5 presents the results of these experiments. The training data consist of phone numbers with the same prefix of 'Contact me at ***'. We observe privacy risk reduction across all prompts, demonstrating the robustness of DEPN to prompt.

## 5.4 Analysis on the Cost-Effectiveness of DEPN

In this subsection we discuss the limitation of DEPN, specifically its dependency on the amount of private data to be erased. We conducted an experiment where we used 1,000 private data instances, each containing phone numbers, extracted from our training dataset. DEPN was applied onto the BERT-base model to erase private information. Experi-

ment results are shown in Table 3. As the amount of private data increases, more neurons need to be edited to achieve better privacy protection, and the performance of the model drops significantly. Furthermore, it becomes apparent that, with the escalation of private data volume, the reduction in privacy risks gradually diminishes. These observations indicate that DEPN excels in remediating language models when dealing with a small number of data leaks, but exhibits weak performance when confronted with a large batch of private data.

## 6 Related Work

**Model Editing** To edit incorrect or undesirable information captured in LLMs, a variety of model editing approaches have been proposed, which can be categorized into four strategies. First, the Constrained Fine-tuning strategy (Zhu et al., 2020) updates LLMs specifically for the target knowledge, allowing precise modification. Second, the Memory-based Editing strategy (Mitchell et al., 2022; Dong et al., 2022) maintains a knowledge cache that stores new information to replace undesirable predictions. Third, the Meta-learning-based Editing strategy (De Cao et al., 2021; Mitchell et al., 2021) introduces editable training based on meta-learning, training model parameters to accommodate editing. Lastly, the Locating and Editing strategy (Geva et al., 2021; Meng et al., 2022; Dai et al., 2022) assumes that knowledge is locally stored in LLMs. This strategy locates specific parameters associated with the knowledge and directly edits parameters to perform editing.

**Privacy Protection**  To address privacy risks in NLP models, various privacy-preserving methods have been proposed, which can be categorized into three main stages of application (Guo et al., 2022; Sousa and Kern, 2023): data processing stage, pre-training and/or fine-tuning stage, and post-processing stage. In the data processing stage, methods involve removing or replacing sensitive information in the original data (Liu et al., 2017; El Emam et al., 2009; Zhou et al., 2008; García-Pablos et al., 2020). In the pre-training or fine-tuning stage, data privacy can be protected by modifying the model training process. One approach is differential privacy stochastic gradient descent (DP-SGD) (Li et al., 2021; Hoory et al., 2021), which introduces noise into the clipped gradient to reduce the distinction between gradients and prevent memorization of training data. Another method is adversarial training (Plant et al., 2021; Coavoux et al., 2018), which constrains the model's learning of private information through adversarial training techniques. However, methods used in the data processing stage and in the pre-training or fine-tuning stage are not applicable if the privacy leakage is discovered after the model training is completed. Methods used in the post-processing stage focus on making trained models forget specific data or alter specific parameters to safeguard hidden private information (Bourtoule et al., 2021; Gupta et al., 2021; Neel et al., 2020). These methods are often with high computational cost and cannot be easily applied to large models. In contrast, proposed DEPN can achieve the protection of private information in the post-processing stage with a small computational overhead.

# 7   Conclusion

In this paper, we have presented a privacy neuron detecting and editing framework DEPN to address privacy leakage risks in pretrained language models. Through the privacy neuron detector based on the privacy attribution scoring method, we accurately detect risky neurons associated with private information. The privacy neuron editor effectively eliminates model memorization of private data. Experimental results and in-depth analyses demonstrate the ability of DEPN to reduce privacy risks efficiently without degrading model performance. Our work explores a novel approach to privacy protection and contributes to model de-memorization in the post-processing stage.

**Limitations**  Our current study still has two limitations. First, although we propose a method to process private data in batches, we find that too many instances in a batch will reduce the effect of memorization erasure. Second, we use a few types of private information in our experiments due to the limited availability of datasets containing private information. We would like to collect more available datasets for our framework in the future.

**Ethical Statement**  In this paper, we use the Enron dataset to evaluate the privacy-preserving effect of DEPN. This dataset consists of employee emails that were publicly disclosed during Enron's legal investigation by the Federal Energy Regulatory Commission. Since the data comes from real persons, we masked sensitive information such as specific names and phone numbers in this paper.

# Acknowledgements

The work was partially supported by the research collaboration project between Tianjin University and ByteDance(PJ20210625900030) and Zhejiang Lab (No. 2022KH0AB01). We would like to thank the anonymous reviewers for their insightful comments.

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

# A Appendix

## A.1 Training Details

For **BERT-base** fine-tuning, we set the hyperparameters as follows: 20 training epochs, a learning rate of 3e-5 with linear warm-up, and a batch size of 16. We fine-tuned **BERT-base** on the Enron dataset using the Masked Language Modeling task to simulate training on datasets containing privacy information. Additionally, we pretrained smaller (*layer=4, hidden size=512, intermediate size=2048*) (Bhargava et al., 2021) and larger (*layer=24, hidden size=1024, intermediate size=4096*) **BERT** models [3] to compare the performance of privacy erasure at different model scales.

## A.2 Effect of the Frequency of Privacy Data Ocurrence

We also examined the influence of the frequency of privacy data ocurrence in the training set on DEPN. As shown in Table 6, phone numbers with an ocurrence frequency greater than 10 exhibit higher exposure compared to those with a frequency less than 10, indicating a higher risk of leakage. However, after erasure, the exposure of phone numbers with a frequency greater than 10 is reduced by 32.65%, while the exposure of phone numbers with a frequency less than 10 is reduced by 22.58%. These results suggest that our method effectively reduces exposure for both high-frequency and low-frequency phone numbers, mitigating the risk of privacy leakage.

| Frequency | Original Exposure | Exposure |
|-----------|-------------------|----------|
| >=10      | 23.15             | 15.59    |
| <10       | 8.90              | 6.89     |

Table 6: Frequency of privacy data ocurrence make exposure different.

## A.3 Effect of Training Time

Figure 4 illustrates the changes in exposure of phone number data before and after erasing privacy neurons in models with different training epochs. We conducted experiments using 20 different phone numbers and averaged the final results. The blue line represents the exposure of phone numbers before privacy neuron erasing. The blue line initially remains low but exhibits a significant surge after the 10-th epoch, indicating that models

---

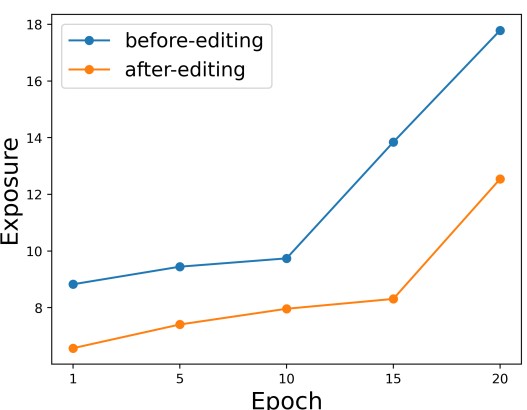

Figure 4: Comparison of privacy leakage risk reduction at different training epochs.

with longer training time have a more pronounced memorization of the training data. Additionally, the yellow line represents the exposure of phone numbers after privacy neuron erasing. The widening gap between the two lines indicates that as the model's memorization becomes more apparent, the proposed DEPN becomes more effective in privacy preservation.

## A.4 The Judgement of the Memorization

In our experiment, we specifically identify the private data memorized by the language model from the training dataset. To assess whether the model has memorized private data, we employ the context of private information as the input to the language model. Subsequently, we calculate the risk of private information leakage and classify the information with a leakage risk exceeding predefined thresholds as having been memorized by the language model. For names, we establish a threshold for memorization as the MRR of less than 1.5. For phone numbers, we employ the Exposure values exceeding 15 as memorization.

---

[3] https://huggingface.co/BERT-large-uncased