# OpenReview forum: "DEPN: Detecting and Editing Privacy Neurons in Pretrained Language Models"
_EMNLP/2023/Conference — EMNLP 2023 Main_

### Official Review · Reviewer_iqTb · 2023-08-04

**Soundness:** 4

**Excitement:**

4: Strong: This paper deepens the understanding of some phenomenon or lowers the barriers to an existing research direction.

**Paper Topic And Main Contributions:**

This paper introduced a privacy preservation method for pre-trained language models, which directly locates "privacy" neurons in the language model and edit these detected neurons by setting the activations to zero. Experiments on a Enron dataset and BERT showed that the method can do privacy preservation without losing too much model performance.

**Questions For The Authors:**

Question A: The privacy information in Enron dataset can be regarded as different entities, why not use accuracy/F1 to evaluate the privacy detection performance?

**Reasons To Accept:**

* A simple and direct method for privacy preservation in pre-trained language models.
* Well written paper, easy to follow.


**Reasons To Reject:**

* In the EEnron dataset, only three types of private information is evaluated, it would be better to use other datasets (e.g. biomedical domain) which contains more types of private information.
* It's a white box method, may not be suitable for black box language model APIs.


**Reproducibility:**

4: Could mostly reproduce the results, but there may be some variation because of sample variance or minor variations in their interpretation of the protocol or method.

**Reviewer Confidence:**

4: Quite sure. I tried to check the important points carefully. It's unlikely, though conceivable, that I missed something that should affect my ratings.

---

> ### Author Rebuttal · Authors · 2023-08-28
>
> Thank you so much for your insightful comments and suggestions. We answer your comments and questions as follows.
>
> - Response to R1:
>
> We agree with you that more different types of privacy information are needed to demonstrate the robustness and applicability of our method. Medical domain is indeed a valuable domain for assessing privacy preservation; however, datasets in this domain often require additional authorizations for use. We have attempted to obtain access to the MIMIC-III clinical dataset, but have not received a response yet. If we finally gain access to such privacy datasets, we are happy to immediately update our experiment results in the next version.
>
> - Response to R2:
>
> Many thanks for this suggestion. Our approach mainly aims at post-processing privacy protection before a language model is officially deployed by its developer.
>
> - Response to Q1:
>
> Thank you for your question. The reason we have not employed ACC/F1 metrics to measure privacy leakage lies in the challenge of accommodating intricate privacy data types. For instance, when dealing with privacy data presented as complete sentences, determining the denominator for accuracy becomes considerably complex.

---

### Official Review · Reviewer_dXCH · 2023-08-05

**Soundness:** 4

**Excitement:**

4: Strong: This paper deepens the understanding of some phenomenon or lowers the barriers to an existing research direction.

**Paper Topic And Main Contributions:**

The paper addresses the problem of privacy leaking in Masked language models and proposes an approach for identifying and editing neurons that can lead to privacy leakage.

**Questions For The Authors:**

Q1. How many samples in the batch can achieve optimal forgetting without hurting the model performance?

**Reasons To Accept:**

The paper addresses a crucial problem in language models. It offers an effective solution to overcome it by balancing between making the model private and does not hurt the general performance of the model by a significant margin.

**Reasons To Reject:**

R1. Even though MLM is necessary and used in many applications, the method should also be employed in the autoregressive language models, which are more used in real-world applications. Also, the problem of memorization is observed and analyzed in many research. Employing it in the autoregressive LM is crucial to prove the significance of the approach. Furthermore, making the LM private usually hurts the general LM performance; the only way to check this in MLM is perplexity. Conversely, autoregressive LM has many downstream benchmarks that can be evaluated to ensure the model's effectiveness.

R2. The paper only addresses limited types of information and, in some cases, formally specified data (phone numbers).

R3. it is mentioned in the limitations section that "many instances in a batch will reduce the effect of memorization erasure" from line 575 to line 578. However, the authors did not analyze this observation and relationship by illustrating when the editing becomes invalid when increasing the batch size.

R4. The paper only compared one baseline method. Comparing the approach with various methods in the literature would be better.

**Reproducibility:**

4: Could mostly reproduce the results, but there may be some variation because of sample variance or minor variations in their interpretation of the protocol or method.

**Reviewer Confidence:**

5: Positive that my evaluation is correct. I read the paper very carefully and I am very familiar with related work.

---

> ### Author Rebuttal · Authors · 2023-08-28
>
> Thank you so much for your insightful comments and suggestions. The following are our answers to your comments and questions.
>
> - Response to R1:
>
> We sincerely appreciate this insightful suggestion. We have indeed employed our method to autoregressive language models and conducted experiments. Due to the page limitation, we haven’t included them in the current version. But we have updated our GitHub repository (https://github.com/Take2try/DEPN) with code and results on GPT-2. The results on GPT-2 suggest that DEPN achieves promising privacy preservation performance on autoregressive models, as observed on BERT. We’ll provide these results and analyses in the new version with additional pages.
>
> - Response to R2:
>
> Due to existing legal and ethical constraints, there is a scarcity of publicly available datasets containing privacy information. As a result, we are only able to conduct experiments on very limited privacy information types available in the public dataset Enron. We are willing to validate DEPN on more privacy information types if more public privacy-related datasets are available in the future.
>
> - Response to R3 & Q1:
>
> This is an exceptionally insightful suggestion. Our experiments indeed demonstrate that erasing a larger batch of samples requires editing more neurons, leading to a decline in model performance. When increasing the batch size of privacy samples while keeping the number of erased neurons constant, the DEPN's ability to erase privacy samples becomes more difficult.
> Ultimately, our findings indicate that maintaining a balance between privacy preservation performance and model efficacy is achievable when the batch size does not exceed 10,000. We’d like to discuss more on this in the next version.
>
> - Response to R4:
>
> Privacy preservation methods typically encompass data preprocessing and model training stages. DEPN, functioning as a post-processing privacy preservation method, holds a unique advantage of offering remediation for already trained models. In this aspect, we do not have many baseline methods to compare, which also works in a post-processing way like ours.

---

### Official Review · Reviewer_xo7t · 2023-08-12

**Soundness:** 4

**Excitement:**

4: Strong: This paper deepens the understanding of some phenomenon or lowers the barriers to an existing research direction.

**Paper Topic And Main Contributions:**

The paper proposes the DEPN (Detecting and Editing Privacy Neurons) method, comprised of a gradient-based privacy neuron detector, an on-off neural privacy editor, and a privacy neuron aggregator for multiple sentences. They show by BERT model on Enron dataset that the proposed method can reduce the detect privacy neurons without hurting the model ability.

**Reasons To Accept:**

1. The first one to apply neuron-based model editing techniques to the privacy issue of language models
2. The method is efficient to reduce the privacy issue while maintaining the model performance successfully

**Reasons To Reject:**

1. Such neuron-level editing method may be limited by the complexity of the issue: once the privacy issue involves lots of different types of information, or involves large amount of data, the number of to-be-edited neurons would be increasingly large, and thus may degrade the model performance. Therefore, the extendability to more complicated privacy issue is questionable.

**Reproducibility:**

4: Could mostly reproduce the results, but there may be some variation because of sample variance or minor variations in their interpretation of the protocol or method.

**Reviewer Confidence:**

2: Willing to defend my evaluation, but it is fairly likely that I missed some details, didn't understand some central points, or can't be sure about the novelty of the work.

---

> ### Author Rebuttal · Authors · 2023-08-28
>
> Thank you so much for your insightful comments and suggestions. We are pleased to answer two key aspects raised in your comments.
>
> - Types of Privacy Information:
>
> We concur with you regarding the increased complexity posed by diverse types of privacy information. However, due to existing legal and ethical constraints, publicly available datasets containing privacy information remain relatively scarce. Consequently, our approach centers on addressing the specific privacy information type of individual identities in Enron. Nonetheless, we believe that exploring a broader spectrum of privacy information types is a worthwhile research direction.
>
> - Amount of Privacy Data:
>
> Using DEPN to erase large amount of privacy data requires editing a great number of neurons, which, however, results in rapid deterioration of model performance. Large-scale neuron editing is also very challenging for model editing. We’d like to continue our work in this direction and explore machine unlearning methods to precisely edit activation values of privacy neurons. Many thanks for this great suggestion.

---

### Meta-Review · Area_Chair_ebJL · 2023-09-16

**Recommendation:** 5

**Metareview:**

All reviewers concur that the manuscript proposes an effective method to make LLMs private without hurting their general performance. To make the paper even more appealing, the authors are advised to address the following considerations: a) evaluate the cost-effectiveness of the proposed method when extensive neuron must be modified to ensure intricate privacy; b) provide guidance on determining an optimal batch size that balances preserving privacy and maintaining model performance.

---

### Decision · Program_Chairs · 2023-10-07

**Decision:**

Accept-Main

**Comment:**

All reviewers concur that the manuscript proposes an effective method to make LLMs private without hurting their general performance. To make the paper even more appealing, the authors are advised to address the following considerations: a) evaluate the cost-effectiveness of the proposed method when extensive neuron must be modified to ensure intricate privacy; b) provide guidance on determining an optimal batch size that balances preserving privacy and maintaining model performance.